

# Compression of ERA5 meteorological reanalysis data and their application to simulations with the Lagrangian model for Massive Parallel Trajectory Calculations (MPTRAC v2.7)

Farahnaz  Khosrawi[1] and Lars Hoffmann[1]

[1]Jülich Supercomputing Centre, Forschungszentrum Jülich, Jülich, Germany

**Correspondence:** Farahnaz Khosrawi (f.khosrawi@fz-juelich.de)

**Abstract.** Computer performance has increased immensely in recent years, but the ability to store data has only increased slightly. The storage requirements for the current version of the ERA5 meteorological reanalysis data provided by the European Centre for Medium-Range Weather Forecasts (ECMWF) have increased by a factor of $\sim 80$ compared to its predecessor ERA-Interim. This presents scientists with major challenges, especially if data covering several decades is to be stored on local computer systems. Accordingly, many compression methods have been developed in recent years with which data can be stored either lossless or lossy. Here we test three of these methods: two lossy compression methods, ZFP and Layer Packing (PCK), and the lossless compressor ZStandard (ZSTD). We investigate how the use of these compressed data affects the results of Lagrangian air parcel trajectory calculations with the Lagrangian model for Massive-Parallel Trajectory Calculations (MPTRAC). We analyzed 10-day forward trajectories that were globally distributed over the free troposphere and stratosphere. The largest transport deviations (up to 1600 km) were derived when using ZFP with the largest compression (CR=25). Using a less strong compression we could reduce the transport deviation (up to 100 km) and still obtain a significant compression (CR=7). Since ZSTD is a lossless compressor, we derive no transport deviations when using these compressed files for trajectory calculations, but do not reduce the use of disk space significantly using this compressor (reduction of $\sim 30\%$, CR=1.5). The best compromise concerning compression efficiency and transport deviations is derived with the layer packing method PCK. The data is compressed by about $50\%$ (CR=2) but horizontal transport deviations do not exceed 40 km. Thus, our study shows that the PCK compression method would be valuable for application in atmospheric sciences and that with compression of the ERA5 meteorological reanalyses data one can overcome the challenges of high demand of disk space from this data set.

## 1 Introduction

Atmospheric models depend on meteorological input data and the current ERA5 reanalysis product (Hersbach et al., 2020) poses a significant challenge to the atmospheric science community. The increased spatial and temporal resolution and other improvements in the representation of geophysical processes as e.g. improvements in the forecast model and the higher amount of observational data that has been assimilated into the model comes with a significant increase in the needed computing resources and storage requirements compared to its predecessor ERA-Interim (Dee et al., 2011). For example, the computational time and main memory requirements have increased by a factor of $\sim 10$ and the total disk space required for input data had





increased by a factor of ∼80 for the Lagrangian particle trajectory simulations that were presented in Hoffmann et al. (2019).
While this higher demand is acceptable for air parcel trajectory studies that usually cover short time periods (in the order of
days) this poses a significant problem for global simulations or climate simulations that cover several years or even decades.

One way to overcome the problem with the increased disk space, increased memory and computational time (and is com-
monly used) is to downsample the data (e.g. Hoffmann et al., 2019; Bourguet and Linz, 2022; Clemens et al., 2024a, b). The
comparison performed by Hoffmann et al. (2019) for Lagrangian air parcel trajectories showed that there still is an improve-
ment compared to the simulations with ERA-Interim. Also for global simulations this practice has been proven valuable (e.g.
Kirner et al., 2015; Khosrawi et al., 2017, 2018; Ploeger et al., 2021). However, this usually comes with a loss in accuracy,
especially in a loss in the fine-scale structure of the data (see Fig. 1 top panels). Thus, in cases where the fine-scale structure
of the data is of importance, this is no preferable solution. Another, more efficient and accurate, solution would be to compress
the data (see Fig. 1 bottom panels). Techniques such as data compression can significantly improve power efficiency by reduc-
ing both network and storage requirements. Therefore, the reduction of data volumes is an effective way to minimize energy
consumption in storage systems (Alforov et al., 2019).

Compression algorithms can be described either as being lossless or lossy. In the case of lossless compression algorithms, the
disk space is reduced by replacing repeated sequences with a smaller, unique identifier. Thus, an entire dataset can be restored,
when decompressed, without alteration of the original dataset (hence the name, lossless). Lossy algorithms, however, reduce
disk space by manipulating numeric arrays, i.e. the mantissa of individual floating-point numbers. Typically, insignificant bits
are replaced with a sequence of zeros or ones that result in data compression at the cost of numerical inconsistencies between
the original and the compressed datasets (Walters and Wong, 2023).

In the past decade many different lossy and lossless compression methods have been developed. A detailed overview over
current available compression methods is provided in Duwe et al. (2020). Although important in many applications, lossless
methods rarely achieve more than $1.5\times$ compression on double-precision data, and therefore have only limited impact on
bandwidth reduction (Lindstrom, 2014). Lossy methods on the contrary can achieve very high compression ratios (Klöwer
et al., 2021). Some examples for lossless methods that are most public, reliable and widely used are e.g. bzip2 and gzip
(Walters and Wong, 2023). Another example for a lossless compression method, that is similar to gzip, is the ZStandard
(ZSTD, https://datatracker.ietf.org/doc/html/rfc1952).

A further lossless compression format that is very efficient in compressing data is the GRIB format (https://www.dwd.de/
DE/derdwd/it/_functions/Teasergroup/grib_de_bf.html). However, the GRIB format has the disadvantage that it only allows to
store metadata that match predefined tables. Thus, it is not as general or self-describing format as the, in the atmospheric science
community widely used, HDF5 or netCDF4 formats (Silver and Zender, 2017). In order to be as flexible as with netCDF4 and
HDF5 and improving the compression that are typically achieved with these so that they are comparable to GRIB2, Silver
and Zender (2017) introduced a new compression method called "layer-packing". Layer packing simultaneously exploits lossy
linear scaling and lossless compression. Another example for a lossy compression method with which high compression ratios
can be achieved is ZFP (Lindstrom, 2014).



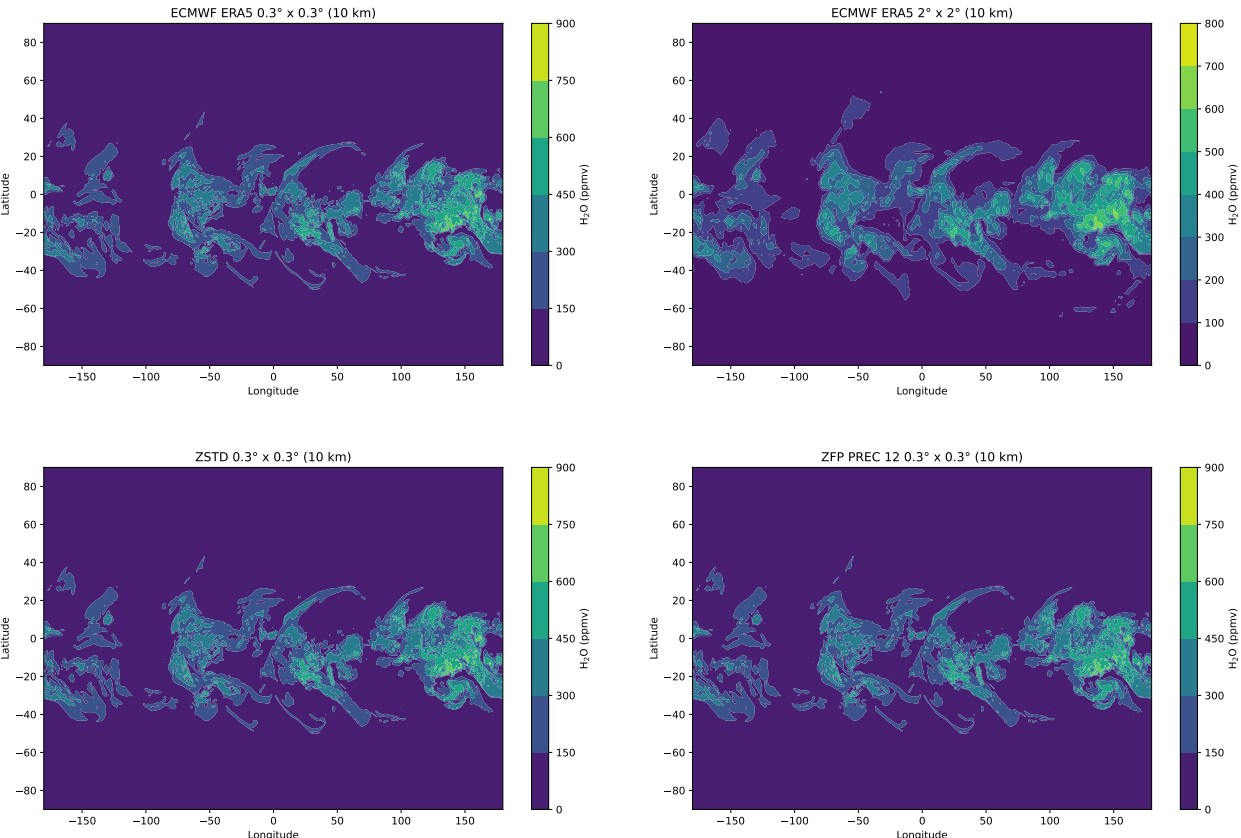

**Figure 1.** 2-d output extracted with the module `met_map` for the 3-d data stored in the ERA5 netCDF files. Shown is $H_2O$ (in ppmv) at 10 km on 8 January 2017 at 17:00 UTC from ERA5 (top left), ERA5 downsampled to $2° \times 2°$ (top right), from ERA5 compressed with ZSTD (bottom left) and ERA5 compressed with ZFP12 (bottom right). Note, the color bar range for the upper right panel is slightly different than for the other three panels.

Geoscientific data as e.g reanalysis data from ECMWF have increased immensely in size and their application in full res-
olution has become quite challenging for users. In this study, we use the lossy compression methods ZFP and PCK and the lossless compression method ZSTD to compress ERA5 meteorological reanalyses files. We compress and test the application of these files on air parcel trajectory calculations with the Massive Parallel Trajectory Calculations (MPTRAC) model. The MPTRAC model has been developed to analyse transport processes in the free troposphere and stratosphere (Hoffmann et al., 2016, 2022). Several case studies have been performed with MPTRAC, e.g. on the long-range transport of $SO_2$ from volcanic 65 eruptions (e.g. Wu et al., 2018; Liu et al., 2023). We assess the accuracy of the air parcel trajectories based on their horizontal and vertical transport deviations from the trajectories calculated based on the original data files stored in the netcdf format. To assess the quality and efficiency of the conventional data compression techniques themselves we assess the speed for converting



and reading the data as well as the mean error, the root-mean squared error and the correlation coefficient between compressed and original ERA5 netCDF data files.

## 2 Data and Method

### 2.1 Meteorological data

#### 2.1.1 ERA5 reanalysis data

ERA5 is the fifth generation ECMWF atmospheric reanalysis of the global climate, produced by the Copernicus Climate Change Service (C3S) at ECMWF. ERA5 provides hourly estimates of a large number of atmospheric, land and oceanic climate
variables over the period from January 1940 to present (Hersbach et al., 2020). The data has been retrieved in the GRIB format from the ECMWF data server on a $0.3° \times 0.3°$ ($30 \times 30$ km) horizontal grid ($T_L639$) and 137 vertical levels covering the atmosphere from the ground to up to 0.01 hPa (about 80 km altitude) with a temporal resolution of 1 h. Meteorological variables that have been downloaded are the following: geopotential ($z$), temperature ($T$), specific humidity ($q$), vertical velocity ($w$), surface pressure ($sp$), zonal wind ($u$), meridional wind ($v$), ozone ($O_3$), specific liquid cloud water content ($SLWC$), specific
ice water content (CIWC), zonal wind at 10 m ($u_{10m}$), meridional wind at 10 m ($v_{10m}$) and temperature at 2 m ($T_{2m}$). The GRIB files that were downloaded from the ECMWF data server were then converted to netcdf for the use with MPTRAC.

### 2.2 Compression methods

#### 2.2.1 ZSTD

ZStandard, or ZSTD as short version, is a fast lossless compression algorithm, targeting real-time compression scenarios at zlib-
level and better compression ratios. ZStandard is similar to gzip (https://datatracker.ietf.org/doc/html/rfc1952) and provides high compression ratios. It's backed by a very fast entropy stage, provided by the Huff0 and FSE library (https://github.com/Cyan4973/FiniteStateEntropy). ZSTD also offers a special mode for small data, called dictionary compression. The reference library offers a very wide range of speed/compression trade-off, and is backed by an extremely fast decoder (https://facebook.github.io/zstd/#other-languages). The Zstandard library is provided as open source software and is published as IETF RFC
8878 (https://datatracker.ietf.org/doc/html/rfc8878).

#### 2.2.2 PCK

The compression method called "layer packing" has been introduced by Silver and Zender (2017) and was developed for large netCDF datasets. This compression method simultaneously exploits lossy linear scaling and lossless compression and is most applicable to datasets where the value range of the variables varies strongly along one (or several) dimension, as it is the case
for e.g. mixing ratios of atmospheric constituents. Layer-packing stores arrays (instead of a scalar pair) of scale and offset parameters. The motivation for Silver and Zender (2017) in developing this method was to improve the lossy compression





ratios typically achieved with HDF5 and netCDF4, so that these are more comparable with the high compression achieved by GRIB2. While records in GRIB2 are strictly two-dimensional and only a limited set of metadata is allowed, they achieve a high compression efficiency, built upon JPEG compression method. netCDF4 and HDF5 on the other hand, provide a highly

flexible framework, allowing attributes, groups and user-defined-data types. The layer-packing method from Silver and Zender (2017) combines the desirable features from both, GRIB2 and HDF5/netCDF4.

### 2.2.3 ZFP

ZFP is a compression scheme for 3-D double-precision and floating-point data and was inspired by ideas developed for texture compression of 2-D image data (Lindstrom, 2014). Using this compression scheme, as in most texture compression formats,

the 3-D array is divided into small, fixed size blocks of dimensions $4 \times 4 \times 4$ that are each stored using the same, user-specified number of bits, and which can be accessed entirely independently. At a high level, the method compresses a block by performing the following five steps. First, the values in a block are aligned to a common exponent. Second, the floating-point values are converted to a fixed-point representation. Third, an orthogonal block transform is applied to decorrelate the values. Fourth, the transform coefficients are ordered by expected magnitude and fifth, the resulting coefficients are encoded one "bit

plane" at a time (Lindstrom, 2014). A detailed description of ZFP can be found in Lindstrom et al. (2025) and a detailed error analysis for floating-point data in Diffenderfer et al. (2019).

### 2.3 Trajectory simulations and data compression with MPTRAC

### 2.3.1 MPTRAC model

For the calculation of air parcel trajectories and the compression of the ERA5 data files the Lagrangian model for Massive-

Parallel Trajectory Calculation (MPTRAC) is used (Hoffmann et al., 2016). MPTRAC calculates air parcel trajectories using 4-D linear interpolation of given wind fields and the explicit midpoint method for numerical integration of the kinematic equations of motion (Rößler et al., 2018). Mesoscale diffusion and subgrid-scale wind fluctuations are simulated using a Langevin equation to add stochastic perturbations to the trajectories, closely following the approach applied in the FLEXible PARTicle (FLEXPART) dispersion model (Stohl et al., 2005; Pisso et al., 2019).

There are several modules implemented in MPTRAC that can be used to additionally simulate convection, sedimentation, radioactive decay, hydroxyl chemistry, dry deposition, and wet deposition along the trajectories (Hoffmann et al., 2022). MPTRAC can be run either on a single workstation or since it features a message passing interface (MPI)–open multi-processing (OpenMP)–open accelerators (OpenACC) hybrid parallelization also on high-performance computing (HPC) systems. On HPC systems simulations can be accelerated by using graphics processing units (GPUs) (Hoffmann et al., 2022, 2024b).

Since the development and first release of MPTRAC in 2016 the model has been applied for several case studies, e.g. concerning the transport of $SO_2$ from volcanic eruptions (e.g. Liu et al. (2023) and references therein). The quality of the air parcel trajectories were evaluated by comparison to super-pressure balloon observations for the Antarctic lower stratosphere (Hoff-




mann et al., 2017). Trajectory errors caused by different numerical integration schemes used within the MPTRAC advection module were evaluated by Rößler et al. (2018).

### 2.3.2 Simulation set-up and MPTRAC modules

Here, we used MPTRAC for the compression of the ERA5 data and for trajectory simulations. We have used MPTRAC v2.7 (Hoffmann et al., 2024a) for the trajectory simulations, but the data were compressed with former versions of MPTRAC (v2.5 and v2.4). We simulated 10-day forward trajectories starting on 1 January 2017. We used $10^6$ air parcels that have been globally distributed over the altitude range of 2-48 km. Thus, we calculated $10^6$ air parcel trajectories for each data set considered. In our simulation diffusion has been switched off so that deviations between trajectories are related to differences between the winds and vertical velocities of the compressed and uncompressed meteorological data files and not to random perturbations added to the trajectories. To compress the ERA5 netcdf data files we used the MPTRAC module `met_conv`. Using this module, we read the original ERA5 netcdf files, conduct the pre-processing of the meteorological data to add additional diagnostic variables such as geopotential height, potential vorticity, etc. and compress the data using the respective compression libraries. The new data is then written out either in a plain binary C format without compression or as compressed binary file, with the suffix .bin or of the respective compression method in the file name, respectively. These C binary files can then easily be read in by MPTRAC and thus used as input data for the trajectory calculations.

The PCK library is used as is, without any specifications by the user. ZSTD is used in MPTRAC with the compression level "3" as default, which is also the default compression level of the ZSTD library and corresponds to a fast compression with a moderate compression rate. PCK is used as standard compression method in the current MPTRAC version, ZFP and ZSTD need to be enabled by specifying ZSTD=1 and ZFP=1 in the compilation. Thus, instead of compiling the model with just calling `make`, one needs to compile with `make ZSTD=1 ZFP=1`. For ZFP PRECISION and TOLERANCE can be defined by the user. With the precision the relative accuracy is given by the number of significant bits while with the tolerance the maximum absolute error tolerance allowed is given (i.e. the absolute difference between original and compressed data). We use the tolerance for geopotential height and temperature ($T = 5$ K, $z = 0.5$ km), all other variables are compressed using the precision. The precision can be user defined in MPTRAC by setting the respective control parameter. That is done in this study. We perform several different compressions with ZFP using a PRECISION of 16, 12 and 8. Higher precision results in higher accuracy and therefore lower compression, while lower precision results in higher compression. All 3-d variables have been compressed. These are the following variables: geopotential height ($z$), zonal wind ($u$), meridional wind ($v$), vertical velocity ($w$), temperature ($T$), potential vorticity (PV), ozone ($O_3$), water vapour ($H_2O$), liquid water content (LWC), ice water content (IWC), cloud cover (CC).

To extract from the 3-d files data 2-d data on certain altitude levels we use the MPTRAC module `met_map`. The met_map output is then used to plot the data on maps (see Fig. 1) or for the statistical error analyses (Sect. 3.1). For the selection of single trajectories from the MPTRAC trajectory data output files we use `atm_select` which enables to perform comparisons of single trajectories (Sect. 3.2).



## 2.4 Evaluation metrics

### 2.4.1 Efficiency metrics

To quantify the data reduction of a file F we achieved due to the compression we use the compression ratio (CR) which is defined as: CR=size original data file/size compressed data file

$$\text{CR}(F) = \frac{\text{filesize}(F_{\text{orig}})}{\text{filesize}(F_{\text{compr}})} \tag{1}$$


### 2.4.2 Statistical metrics

To evaluate the compression methods we use the following statistical metrics: mean error (ME), root mean square error (RMSE) and Pearson correlation coefficient (r). The mean error is calculated as follows:

$$\text{ME} = \sum_{i}^{N} x_i - y_i \tag{2}$$

The root-mean square error is calculated by:

$$\text{RMSE} = \sqrt{\frac{1}{N} \sum_{i=1}^{N} (|x_i - y_i|)^2} = \sqrt{\frac{1}{N} \sum_{i=1}^{N} (e_{abs_i})^2} \tag{3}$$

where $e_{abs_i}$ is the absolute difference between each data point $x_i$ and $y_i$ of the data sets $x$ and $y$, which are in our case the compressed and reference data, respectively.

Additionally to the RMSE, the normalised RMSE (NRMSE) is considered, which is defined as follows:

$$\text{NRMSE} = \frac{\text{RMSE}}{R_x} \tag{4}$$


where $R_x$ is the range of $x$, thus $R_x = x_{max} - x_{min}$.

The Pearson correlation coefficient $r$ is defined as:

$$r_{xy} = \frac{\sum_{i}^{N} (x_i - \bar{x})(y_i - \bar{y})}{\sqrt{\sum_{i}^{N} (x_i - \bar{x})^2 \sum_{i}^{N} (y_i - \bar{y})^2}} \tag{5}$$

where $n$ is the sample size, $x_i$ and $\bar{x}$ are the values and mean of the compressed file and $y_i$ and $\bar{y}$ are the values and mean

of the original file. The Pearson correlation coefficient is a measure for the linear correlation of a dataset. The values of $r$ range between 1 and -1. According to Tao et al. (2019) the compressed data files have a sufficient accuracy when a correlation coefficient of 0.99999 (five nines after the comma) is reached.



## 2.5 Trajectory metrics

For assessing the accuracy of the trajectory simulations, i.e. the difference of two sets of trajectory simulations, various statisti-
cal methods have been proposed. Spatial differences of trajectories are commonly measured in terms of the absolute horizontal
(AHTD) and vertical transport deviation (AVTD), respectively (Kuo et al., 1985; Rolph and Draxler, 1990; Stohl, 1998). AHTD
and AVTD at time t are calculated for two sets of $N$ trajectories each by

$$\text{AHTD}(t) = \frac{1}{N}\sum_{i=1}^{N}\sqrt{[X_i(t) - x_i(t)]^2 + [Y_i(t) - y_i(t)]^2} \tag{6}$$

$$\text{AVTD}(t) = \frac{1}{N}\sum_{i=1}^{N}|Z_i(t) - z_i(t)| \tag{7}$$

where $X_i(t)$, $Y_i(t)$ and $Z_i(t)$ as well as $x_i(t)$, $y_i(t)$ and $z_i(t)$ refer to the particle coordinates of the two sets of trajectories, in
our case the particle coordinates of the compressed and reference trajectories, respectively. Relative horizontal transport devi-
ations (RHTDs) and relative vertical transport deviations (RVTDs) are calculated by dividing the absolute transport deviations
by the horizontal or vertical path lengths of the trajectories, respectively (Rößler et al., 2018; Hoffmann et al., 2019, 2022).

## 3 Results

### 3.1 Comparison of compression methods

In the following the three compression methods ZSTD, PCK and ZFP for compressing 1 month of ERA5 data (January 2017)
are compared. Table 1 shows the size of 1 file and files for 1 month from ERA5 (as netcdf) and converted to a C binary
format (bin) as well as compressed with ZSTD, PCK and ZFP. With ZFP we performed compressions using a precision of
16 (corresponding to double precision), 12 and 8 (corresponding to single precision). We use the bin files as reference for
estimating the compression ratio since MPTRAC does some post-processing after reading the data where additional variables
are calculated and stored. Comparing ZSTD, PCK and ZFP16, the highest compression ratios and thus lowest file sizes are
derived with ZFP16. With PCK a 50 % reduction (CR=2) can be achieved. The lowest compression ratio (CR=1.5) is reached
with ZSTD (reduction of 30 %), but since this is, compared to the other two methods, a lossless compression, there will be no
loss in accuracy. Thus, the mean error and RMSE are equal zero for ZSTD. Using ZFP with a precision of 12 or 8 even higher
compression ratios can be reached (CR of 12 and 25, respectively) reducing the size of 1 file from 4.2 GB to only 164 MB for
ZFP8. However, since this is a lossy compression technique this high compression ratios will definitely come with a substantial
loss in accuracy.

In Fig. 2 and 3 the RMSE and NRMSE are shown as bar charts for all variables considered/compressed and for six altitudes
(2, 5, 10, 20, 30 and 40 km). The NRMSE is the normalised RMSE (thus, dividing the RMSE by the value range of the
respective meteorological variable). The NRMSE allows to compare the errors of the variables with each other and between



**Table 1.** Comparison of the size of one file, size of one month of data, compression ratio and compression type for the compression methods used in this study.

|  | ERA5 nc | Bin | ZSTD | PCK | ZFP16 | ZFP12 | ZFP8 |
|---|---|---|---|---|---|---|---|
| Size (1 file) | 3.0 GB | 4.2 GB | 2.8 GB | 1.9 GB | 571 MB | 325 MB | 164 MB |
| Size (1 month) | 2.2 TB | 3.0 TB | 2.0 TB | 1.4 TB | 416 GB | 236 GB | 119 GB |
| Compression ratio (1 month) | — | — | 1.5 | 2 | 7 | 12 | 25 |
| Compression type | — | — | lossless | (lossy)[1] | lossy | lossy | lossy |

[1] PCK applies lossy linear scaling, but lossless compression

the altitudes (which is not possible with the RMSE itself due to the different units and value ranges of the variables). For all variables the RMSEs and NRMSEs are getting larger as larger the compression ratio is. Thus, the lowest RMSEs and NRMEs are found for PCK (CR=2) and the highest for ZFP8 (CR=25). Note, for $T$ the RMSE and NRMSE, respectively, is the same for the three ZFP compressions since here instead of the precision the tolerance was used. Differences between the six altitudes

215 for the respective compression method are low as can be seen from the NRMSE (Fig. 3). Differences of NRMSE between the different altitudes are for the stratospheric altitudes (20, 30 and 40 km) somewhat lower than for the tropospheric altitudes (2, 5, and 10 km). For PCK the NRMSE is almost the same for all variables while for ZFP the NRMSE is somewhat lower for $w$ and for $T$, $H_2O$ and $O_3$ somewhat higher than for the other variables.

To asses if the quality of the meteorological data after compression is sufficient, we also consider the correlation coefficient.

220 According to Tao et al. (2019) a sufficient accuracy is reached when the correlation coefficient is equal or greater than 0.99999 (five nines after the comma). Table 2 shows the correlation coefficient for all variables accessed at 10 km altitude (other altitudes not shown since there are no significant differences between the altitude levels, these can be found in Appendix A instead). For ZSTD, PCK and ZFP16 the desired correlation coefficient is reached for all variables, except for temperature when ZFP16 is

**Table 2.** Correlation coefficient for the meteorological variables that have been compressed (at 10 km).

|  | ZSTD | PCK | ZFP16 | ZFP12 | ZFP8 |
|---|---|---|---|---|---|
| U | 1.0 | 0.99999 | 0.99999 | 0.99999 | 0.99968 |
| V | 1.0 | 0.99999 | 0.99999 | 0.99999 | 0.99974 |
| W | 0.99999 | 0.99999 | 0.99999 | 0.99997 | 0.99782 |
| T | 0.99999 | 0.99999 | 0.99995 | 0.99994 | 0.99940 |
| $H_2O$ | 0.99999 | 0.99999 | 0.99999 | 0.99997 | 0.99797 |
| $O_3$ | 1.0 | 0.99999 | 0.99999 | 0.99998 | 0.99929 |
| LWC | 0.99999 | 0.99999 | 0.99999 | 0.99965 | 0.93567 |
| IWC | 1.0 | 0.99999 | 0.99999 | 0.99997 | 0.99629 |



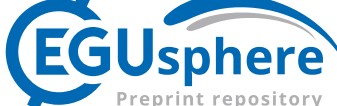

**Figure 2.** Bar chart showing the RMSE at 2 km, 5 km, 10 km, 20 km, 30 km and 40 km for the 3-d variables that have been compressed and all compression methods used in this study (note, for ZSTD the RMSE is zero). The units are the units of the respective variables. These are as follows: $u$ (m s$^{-1}$), $v$ (m s$^{-1}$), $w$ (hPa s$^{-1}$), $T$ (K), H$_2$O (ppmv), O$_3$ (ppbv), LWC (kg kg$^{-1}$), IWC (kg kg$^{-1}$).





**Figure 3.** Same as Fig. 2, but for the NRMSE (unitless since this is the normalised error).

used. For this variable we used the tolerance instead of the precision and it may be that the chosen value was not optimal. Tests
225    using the precision only or swapping the used values of tolerance and precision improved the correlation coefficient for $T$ and
also here five nines after the comma were derived.



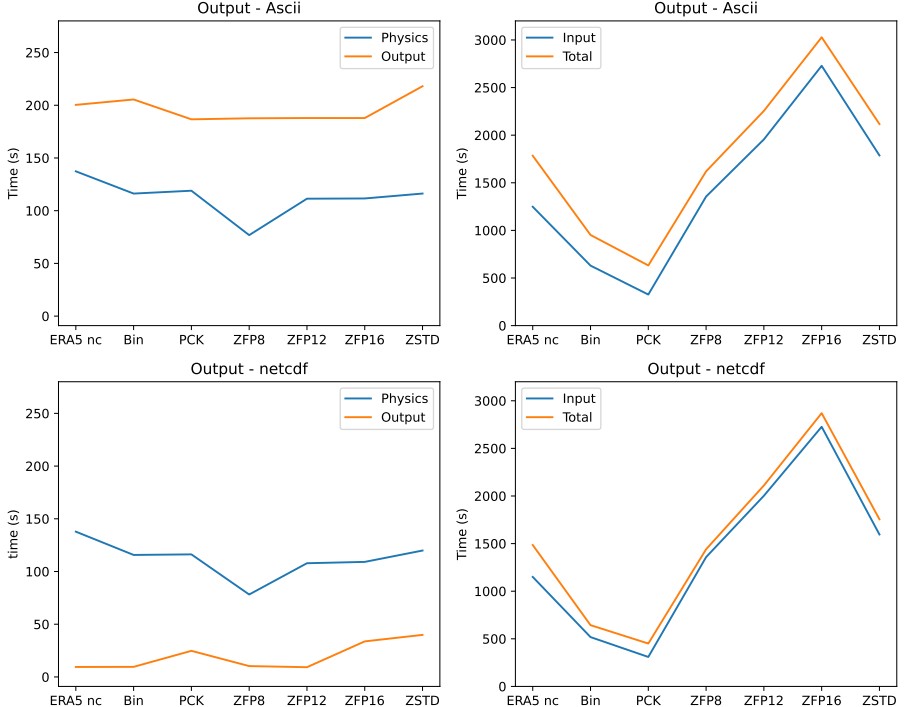

**Figure 4.** MPTRAC runtime comparison using the ERA5 netcdf files, the bin files and the compressed PCK, ZSTD and ZFP files for the 10-day forward trajectory calculations with MPTRAC. Top panels for trajectory output stored as ascii and bottom panels for trajectory output stored as netcdf file.

If the data is stronger compressed the correlation coefficient gets lower. For ZFP12 it is only the fifth digit after the comma, but for ZFP8 several digits after the comma are below the desired nines showing some degradation of the data. How this affects the trajectory calculations with MPTRAC will be assessed in the following section (Sect. 3.2).

## 3.2 Application to Lagrangian trajectory calculations

### 3.2.1 Absolute and relative horizontal and vertical transport deviations

Figure 4 shows a comparison of the required time to read and process the original ERA5 netcdf files and the compressed ERA5 files for trajectory calculations. In our test cases, the total time for reading and processing the data is mostly determined by the time required to read the input data, i.e. the ERA5 data stored in the netcdf or compressed data format. Thereby, the PCK files are read in fastest while the longest time is needed for the ZFP16 files. Reading in the PCK files takes only half the time of the time needed for the netcdf files while it takes 4 times longer for the ZFP16 files. For ZFP8 reading in the data takes approximately the same time as for the ERA5 netcdf files. Storing the trajectory data as ascii file takes longer than storing the data in the netcdf format.



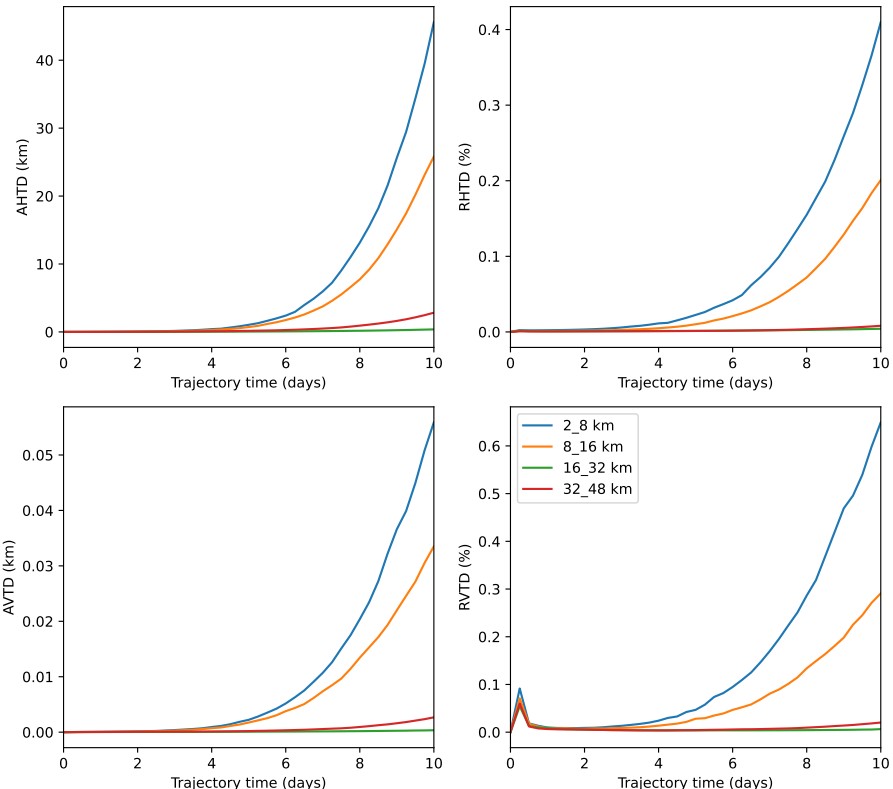

**Figure 5.** Absolute and relative vertical and horizontal transport deviations vs trajectory time for the 10-day forward trajectory calculations with MPTRAC using the meteorological reanalysis files compressed with PCK. The horizontal and vertical transport deviations have been separated into the altitude regions: 2-8 km, 8-16 km, 16-21 km and 32-48 km.

Figures 5 and 6 and Fig. B1 and B2 show the absolute and relative horizontal and vertical transport deviations (AHTD, RHTD, AVTD and RHTD, respectively) for the 10-day trajectories that have been calculated with MPTRAC using the compressed reanalysis data files (PCK, ZFP16, ZFP12 and ZFP8). Note, for ZSTD no transport deviations are shown since this is a lossless compression method and thus there are no transport deviations (these are zero). In the figures (Fig. 5 and 6 and Fig. B1 and B2) the transport deviations are shown separated into altitude regions of 2-8 km, 8-16 km, 16-32 km and 32-48 km. Generally, independent of which compression method is used, the highest horizontal and vertical transport deviations are found at the lowest altitude range of 2-8 km (free troposphere), while the lowest deviations are found at the altitude range of 16-32 km (lower stratosphere). The transport deviations generally increase with trajectory time, reaching their respective horizontal and vertical maximum after 10 days.

Using the PCK compressed files for the trajectory calculation, there are transport deviations of negligible order up to 4 days, but these then start to increase. A maximum horizontal transport deviation of 40 km (0.4%) is reached after 10 days for the lowest altitude range of 2-8 km (Fig. 5). Corresponding vertical transport deviations increase up to 0.06 km (0.6%). Using for





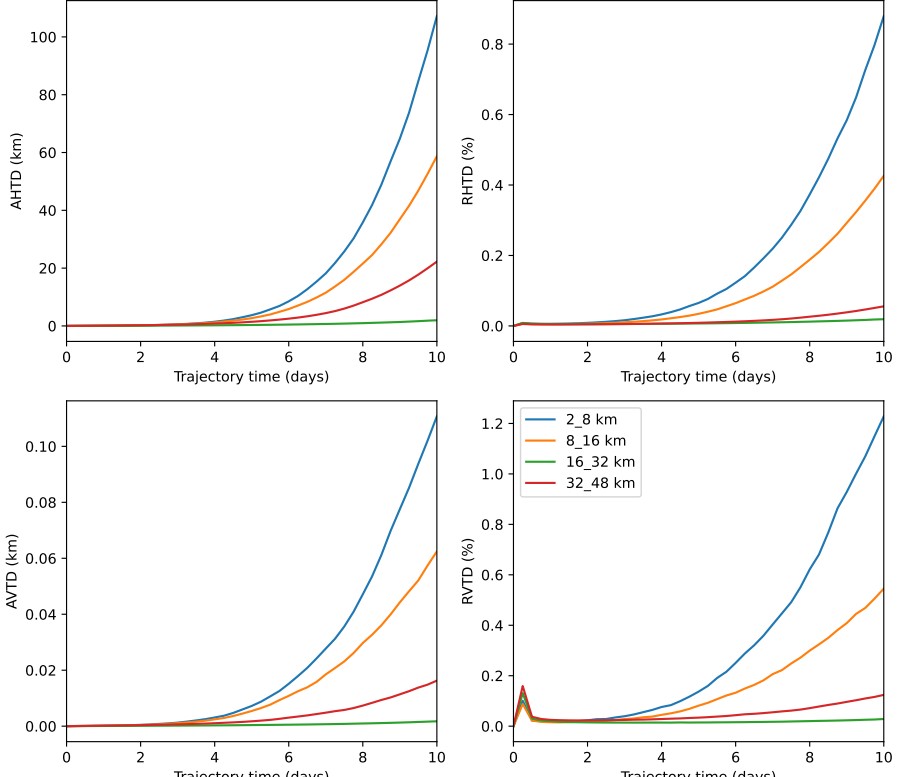

**Figure 6.** Same as Fig. 5 but for ZFP16.

the MPTRAC simulation the files that were compressed with ZFP16 (Fig. 6), the horizontal transport deviations are somewhat higher. Also here, horizontal and vertical transport deviations are quite low/non-existent up to a trajectory time of 4 days and start then to increase, reaching up to 100 km (0.8%) horizontally and up to 0.09 km (1%) vertically.

If the reanalysis data is higher compressed, using ZFP12 and ZFP8, the loss in accuracy of the data is then also reflected in the trajectory calculations (Fig. B1 and B2). Transport deviations start here already to increase from $\sim$ 2 days trajectory time onward and reach horizontal transport deviations of 400 km (3.5%) and 1600 km (0.8%), respectively. Vertical transport deviations increase up to 0.35 km (4%) and 0.8 km (9%), respectively.

Figure 7 shows an example of a trajectory where large deviations (several thousand km) occur. This trajectory has been started in the southern hemispheric troposphere at approximately 8 km altitude (107°W, 62°S). The air mass of reference trajectory that has been calculated using the ERA5 netcdf files, is transported for 7 days eastwards to 118°E, 55°S and then turns back westward and ends at 107°E, 62°S. The trajectories calculated using the compressed PCK and ZFP16 files follow the path of the reference trajectory up to 7 days until the reference trajectory turns eastward. The PCK and ZFP16 trajectories start then to deviate, but remain in the same area. Completely different paths after already 4 days take the trajectories that have been calculated using the data compressed with ZFP8 and ZFP12. These trajectories turn north and end over the Atlantic Ocean





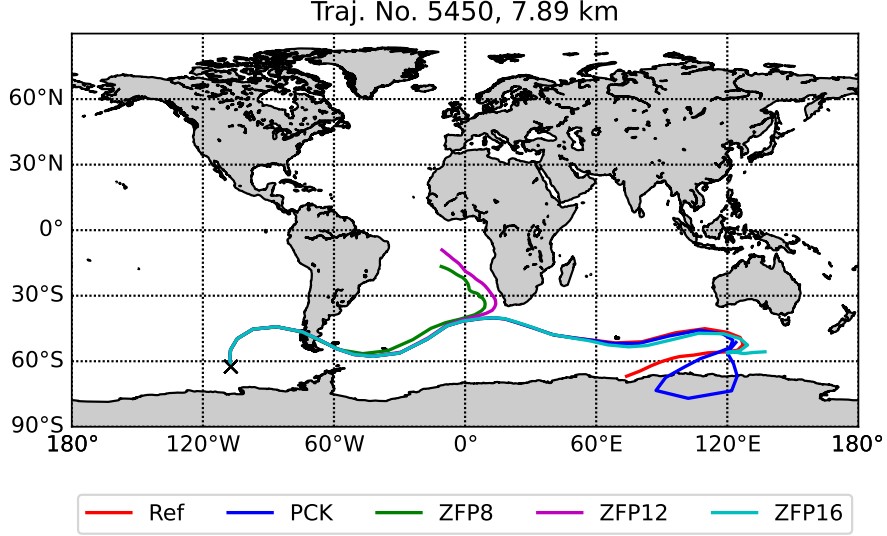

**Figure 7.** A case where using the compressed files for the trajectory calculations results in large deviations in the path of the trajectories calculated with MPTRAC. The reference trajectory (red) is the one that has been calculated using the ERA5 netcdf files. The color coding for the trajectories calculated based on the compressed data files is as follows: PCK (blue), ZFP8 (green), ZFP12 (magenta) and ZFP16 (cyan).

west of Africa. Thus, if with these trajectories air mass transport is investigated, the ones calculated with ZFP8 and ZFP12 may falsify the results (if we assume the reference is the truth) and thus may lead to different conclusions.

### 3.2.2 Statistical analyses of occurrences of large transport deviations

The above discussed example trajectory shows what impact large compression errors may have if these cause large transport deviations. However, the question is how often do these large deviations actually occur, i.e. how many trajectories of the $10^6$ trajectories that we have calculated are actually affected. Another question is whether certain heights at which the trajectories have been started are affected more than others. To answer this, we analysed the trajectory data based on the starting altitudes and their corresponding frequency of maximum deviations. As threshold for the maximum deviations at the trajectory end points we used the maximum deviations we found from the calculations of the AHTDs (Fig. 5 and 6 and Fig. B1 and B2). Thus, for PCK we use a threshold of $> 40$ km, for ZFP16 $> 100$ km, for ZFP12 $> 400$ km and for ZFP8 $> 1600$ km. The results are shown in Fig. 8.

From the trajectories calculated using the PCK data in total 36 833 (3.7%) have deviations greater than 40 km, from the ZFP16 trajectories 51 148 (5.1%) have deviations greater than 100 km, from ZFP12 the trajectories 103 739 (10.4%) have deviations greater than 400 km and from the ZFP8 trajectories 175 171 (17.6%) have deviations greater than 1600 km after 10 days. This means, not only the deviations themselves increase with higher compression (Table 1), but also the frequency with which these maximum deviations occur. For all four trajectory data sets simulated based on the compressed ERA5 data



files, the maximum deviations are lowest or do not occur at start altitudes between 20 and 30 km. The highest frequency of the maximum deviations is found for all trajectory data sets for the troposphere. Quite low maximum deviations are also found for the trajectories started in the upper stratosphere (40-50km) that have been calculated using the the PCK compressed files. Using the ZFP12 compressed files for trajectory calculation the frequency in the upper stratosphere is somewhat higher than

for the ones calculated based on the files compressed with PCK and the ones compressed with ZFP16. For the trajectories calculated based on the ZFP8 compressed data these maximum deviations occur almost as frequent as for the ones started at tropospheric altitudes.

Figure 9 shows the frequency of occurrences of distances of up to 1600 km between the trajectory end points of the trajectories calculated using the ERA5 netcdf files compared to the ones that have been calculated using the compressed data files.

Here, we can assess how often certain distances at the end points occur. For all data sets, the majority of distances at the end points lie below 200 km. For the trajectories calculated using the PCK files this holds for 98.5%. Similar for the ones calculated using the files compressed with ZFP16, here 96.3% have distances lower than 200 km. When with ZFP a lower precision and thus higher compression is used, the percentage is decreasing from 81.5% for ZFP12 to only 35.4% for ZFP8. While for the trajectories calculated using the ZFP12 compressed data files still the majority of trajectories has distances lower than 200 km

at the end points, for the ones calculated using the data files compressed with ZFP8 this is only the case for one-third of the trajectories.

For the trajectories calculated using the PCK and ZFP16 compressed files, the frequency for all other bins is around or below 1%. For the trajectories that were calculated with the ZFP12 and ZFP8 compressed data files, the frequency of distances between 200 and 400 km is 6.4% and 15.6%, respectively, and is between 400 and 600 km 2.8% and 9%, respectively. For the

trajectories calculated with the ZFP8 compressed files, the frequency of higher distances remains above 1% while for ZFP12 the frequency is around or below one percent for the following bins.

In Fig. 10 we assess the frequency of trajectory deviations in dependence of simulation time, i.e after how many days are certain distances between trajectory end points found. As derived from the AHTDs and already used for Fig. 8 we use as threshold values 40 km, 100 km, 400 km and 1600 km. For the trajectories calculated using the PCK compressed files, the

deviations are not exceeding 40 km and this small deviations start to occur after 8 days of simulations time. Up to 8 days almost no deviations are found. For the trajectories calculated using the ZFP16 compressed files a similar behaviour as for the ones using the PCK compressed files is found. Further, these small deviations only occur with a frequency of a few percent. Thus, these trajectories keep a high accuracy over the entire simulation time of 10 days (which is also reflected in the high correlation coefficients discussed in Sect. 3.1).

For higher compressions, as it is the case for ZFP12 and ZFP8, the deviations between the reference trajectories and those calculated using the compressed data start to increase much earlier. Furthermore, these deviations occur with a much higher frequency as already seen in Fig. 8 and 9. For the trajectories calculated using the with ZFP12 compressed data, deviations of >40 km occur already after 3 days and reach a frequency of 30% after 10 days. Deviations of 100 km occur after 4 days and reach a frequency of 20% after 10 days. Deviations of >400 km occur later, namely after 6 and 8 days and have a frequency

of 5% and < 5% percent after 10 days, respectively.





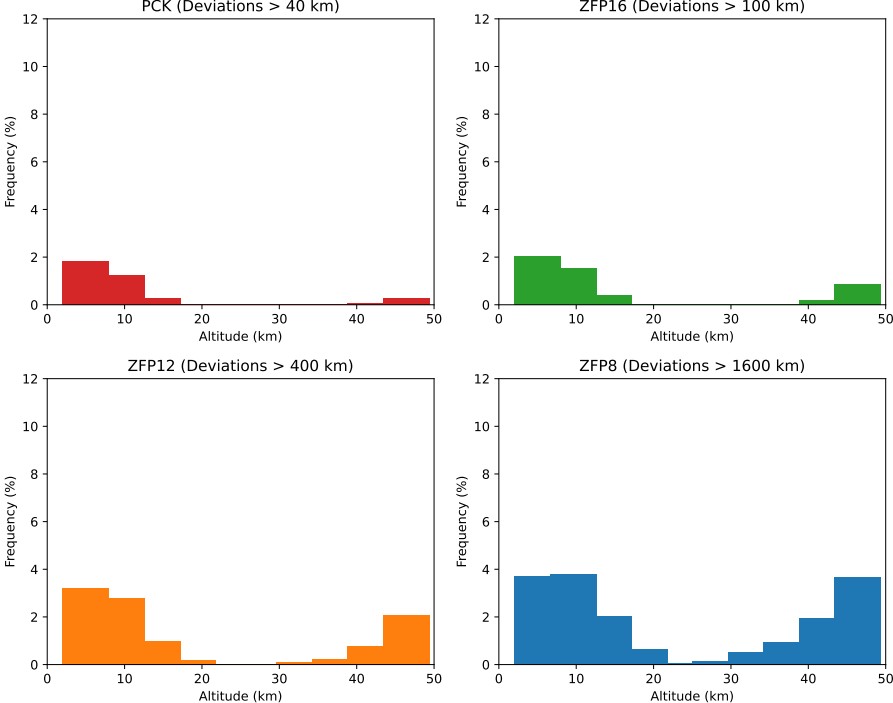

**Figure 8.** Frequency of maximum deviations at the trajectory end points using the threshold of 40 km for PCK (top left), 100 km for ZFP16 (top right), 400 km for ZFP12 (bottom left) and 1600 km for ZFP8 (bottom right), respectively, vs altitude where the trajectories have been started.

If the ERA5 data compressed with ZFP8 are used for the trajectory calculation, deviations >40 km occur immediately and reach a frequency of 80% after 10 days. For larger deviations these occurrence frequencies are as follows: deviations >100 km occur after 1 day and reach up to 60% after 10 days, deviations >400 km occur after 2 days and reach up to 40% and deviations >1600 km occur after 1 day and reach up to 15%.

## 4 Discussion

The advantage of lossy compression is that much larger compression ratios than for lossless compression can be achieved. However, the disadvantage of lossy compression is that there is an unavoidable loss of accuracy. The question is, however, how much loss and thus how large transport deviations in the trajectory calculation are tolerable for atmospheric applications. To get a better overview, we will compare the here derived transport deviation to previous studies where transport deviations have been estimated using MPTRAC for trajectory simulations. There are three studies that have a similar set-up as used in our study (Rößler et al., 2018; Hoffmann et al., 2019, 2022).



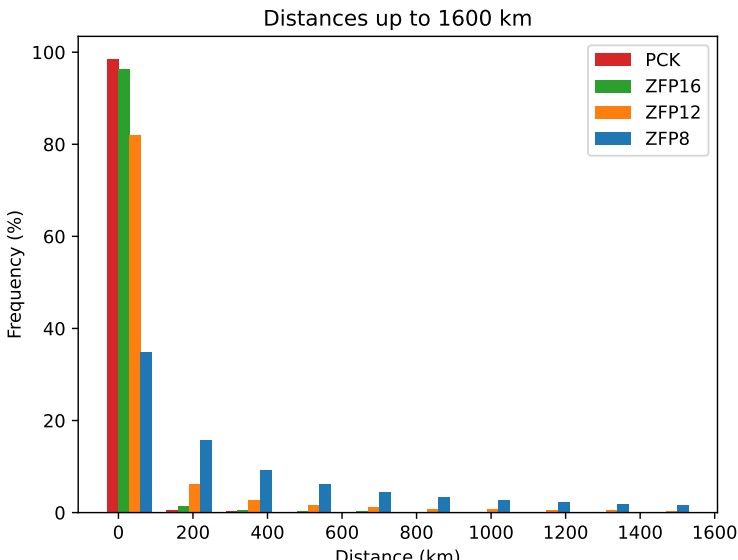

**Figure 9.** Frequency of occurrence of distances up to 1600 km between trajectory end points of the trajectories calculated from the ERA5 netcdf files and from the compressed files. The colors denote the results from the different compression methods that were used to compress the ERA5 data and served as input data set to calculate the trajectories.

Rößler et al. (2018) compared six different numerical integration schemes calculating $500\,000$ globally distributed 10-day trajectories for each integration scheme. Deviations between the schemes were for all schemes low up to 5 days, but then began to rapidly increase for some schemes, resulting in AHTDs of up to 4400 km and AVTDs of 4800 m. However, for the
higher order explicit Runge-Kutta schemes (third and fourth order) deviations of only a few hundred km horizontally and a few hundred m vertically were found after 10 days.

In Hoffmann et al. (2019) differences between transport simulations using ERA5 and ERA-Interim data were quantified. A global set of 10-day forward trajectories for $10^6$ globally distributed air parcels in the free troposphere and stratosphere was analysed. Using ERA5 for trajectory simulation, transport deviations of up to 1400 km horizontally and 800 m vertically were
found due to parameterized diffusion and sub-grid scale wind fluctuations after 10 days. These were about a factor of 2 lower than using ERA-Interim for trajectory calculations.

In another study by Hoffmann et al. (2022) absolute horizontal and vertical transport deviations were calculated in order to quantify the effects of simulated diffusion and sub-grid scale wind fluctuations on 8-day trajectory calculations within the northern hemispheric jet stream. Different parameter choices for diffusion and sub-grid scale wind fluctuations were used.
Depending on the parameter choice, horizontal transport deviations ranged from 100 km to about 1800 km after 10 days. Vertical transport deviations ranged from 100 m to 1 km after 8 days.

Comparing the transport deviations from Rößler et al. (2018); Hoffmann et al. (2019) and Hoffmann et al. (2022) with the ones derived in this study, we find that these lie in the typical range of transport deviations that can be expected when





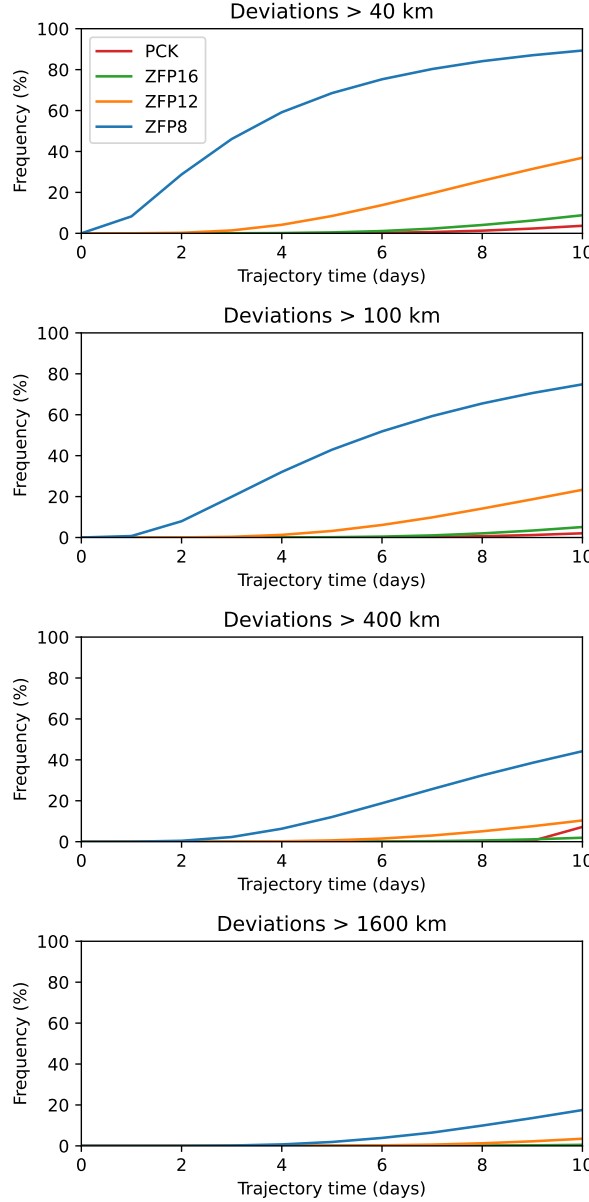

**Figure 10.** Frequency of occurrence of cases where deviations greater 40, 100, 400 and 1600 km are reached in dependence on the trajectory time.

uncertainties from external sources are considered. Nevertheless, the ultimate goal is always to find the best compromise

between accuracy and computational performance. This, however, depends also on for which kind of study the compressed data is used. As shown here for trajectory calculations, we find that the stratosphere is quite insensitive and thus higher compressions are possible.



The length of the trajectory calculation also plays a decisive role. The actual trajectory length (in days) depends on the atmospheric applications. For studies investigating e.g. air mass transport for pollutants, smoke or dust in the troposphere
rather shorter trajectory lengths are considered (∼1-3 days), while in the stratosphere for e.g. studies on transport of long-lived tracers rather longer trajectories are considered (5-80 days or even longer).

For us PCK is the best choice, since it is quite efficient in compressing the ERA5 data (CR=2) while keeping the accuracy of the data. Additionally, the data is quickly read and processed with MPTRAC for trajectory calculations. Nevertheless, in the end the user has to decide which compression method is the one which fits best for their specific needs, but we hope that the
results derived in this comparison will help them to make the right decision.

## 5 Conclusions

To cope with the increasing storage requirements of modern, high resolution meteorological reanalysis data and to be prepared for the upcoming ERA6 reanalyis data, which will have an even higher storage requirement than ERA5, data compression is a promising solution. In this study we applied three compression methods (ZSTD, PCK and ZFP) on the ERA5 reanalysis data
and assessed the impact of using these compressed data sets on trajectory calculations with the Lagrangian model MPTRAC.

The best results in terms of accuracy were obtained with ZSTD since this is a lossless compression method. However, the compression ratio is only 1.5, thus using this compression method only a reduction in data storage of about 30% is achieved. A good compromise between accuracy and compression efficiency is derived with PCK. Using this compression method, we could achieve a compression ratio of 2, thus a reduction in storage of 50%. Absolute horizontal and vertical transport deviations
were quite low. These did not exceed 40 km horizontally and 60 m vertically, which corresponds to relative transport deviations of 0.4% and 0.6%, respectively.

While ZSTD and PCK do not support user-specific settings, ZFP allows the user to set the precision and tolerance. We have tested ZFP with different precision settings to achieve different compression rates. Using a PRECISION of 16 we derived as for PCK a relatively high accuracy, but with a higher compression ratio (CR=7) and with still relatively low absolute horizontal
and vertical transport deviations (up to 0.8 % horizontally and 1 % vertically). If the precision is reduced (PRECISION of 12 or 8) significantly higher compression ratios (up to 25) are achieved, but with a significant loss in accuracy that is reflected in the higher absolute and relative horizontal and vertical transport deviations (deviations up to 13%).

Comparing our results with previous studies, we find that the transport deviations derived here are similar to those derived in the previous studies by Rößler et al. (2018); Hoffmann et al. (2019) and Hoffmann et al. (2022). By performing a statistical
analysis we found that only a minority of trajectories were affected by these large transport deviations. Therefore, we can conclude that all compression methods can be applied to the ERA5 meteorological reanalysis data and that the final decision on which method is the best has to be made by the user based on the specific computational requirements and atmospheric applications. We are confident that the here presented assessment will help users to make the right decision which compression method is the best for their needs. For our needs PCK is the best choice and has thus been implemented in MPTRAC as the
default compression method.



*Code and data availability.* MPTRAC is made available under the terms and conditions of the GNU General Public License (GPL) version 3. The current and former versions of MPTRAC can be downloaded from the github: https://github.com/slcs-jsc/mptrac (Hoffmann et al., 2016, 2022). The ERA5 data can be obtained from the European Centre for Medium-Range Weather Forecasts (ECMWF) Meteorological Archival and Retrieval System (MARS), see https://www.ecmwf.int/en/forecasts/datasets (Hersbach et al., 2020). The code and data for this

study has been published on Zenodo: https://zenodo.org/records/15782664 (Khosrawi and Hoffmann, 2025).

*Author contributions.* FK and LH designed this study. The compression of ERA5 data with PCK was done by LH. FK compressed the ERA5 data with ZFP and ZSTD, performed the data analysis and wrote the manuscript with input from LH.

*Competing interests.* Lars Hoffmann is an editor of Geoscientific model development. Otherwise the authors do not have any conflicts of interest.

*Acknowledgements.* The work presented in this paper was funded by the BMBF project ADAPTEX (FKZ: 16ME0670). We acknowledge the Jülich Supercomputing Centre for providing computing time and storage resources on the JUWELS supercomputer. We would like to thank Rolf Müller and Olaf Stein for helpful discussions and comments on the manuscript. For some text parts deepL Write has been used in order to improve our language.





## Appendix A: Correlation coefficient

**Table A1.** Correlation coefficient for ZSTD at 2, 5, 10, 20, 30 and 40 km altitude.

| Altitude | 2 | 5 | 10 | 20 | 30 | 40 |
|----------|---------|---------|---------|---------|---------|---------|
| U | 0.99999 | 1.0 | 1.0 | 1.0 | 0.99999 | 1.0 |
| V | 1.0 | 0.99999 | 1.0 | 1.0 | 1.0 | 0.99999 |
| W | 1.0 | 0.99999 | 0.99999 | 0.99999 | 0.99999 | 1.0 |
| T | 0.99999 | 0.99999 | 0.99999 | 0.99999 | 0.99999 | 0.99999 |
| $H_2O$ | 0.99999 | 1.0 | 0.99999 | 0.99999 | 1.0 | 1.0 |
| $O_3$ | 0.99999 | 0.99999 | 1.0 | 1.0 | 0.99999 | 0.99999 |
| LWC | 1.0 | 0.99999 | 0.99999 | - | - | - |
| IWC | 0.99999 | 0.99999 | 1.0 | 0.99999 | - | - |

**Table A2.** Correlation coefficient for PCK at 2, 5, 10, 20, 30 and 40 km altitude.

| Altitude | 2 | 5 | 10 | 20 | 30 | 40 |
|----------|---------|---------|---------|---------|---------|---------|
| U | 0.99999 | 0.99999 | 0.99999 | 0.99999 | 0.99999 | 0.99999 |
| V | 0.99999 | 0.99999 | 0.99999 | 0.99999 | 0.99999 | 0.99999 |
| W | 0.99999 | 0.99999 | 0.99999 | 0.99999 | 0.99999 | 0.99999 |
| T | 0.99999 | 0.99999 | 0.99999 | 0.99999 | 0.99999 | 0.99999 |
| $H_2O$ | 0.99999 | 0.99999 | 0.99999 | 0.99999 | 0.99999 | 0.99999 |
| $O_3$ | 0.99999 | 0.99999 | 0.99999 | 0.99999 | 0.99999 | 0.99999 |
| LWC | 0.99999 | 0.99999 | 0.99999 | - | - | - |
| IWC | 0.99999 | 0.99999 | 0.99999 | 0.99999 | - | - |



**Table A3.** Correlation coefficient for ZFP16 at 2, 5, 10, 20, 30 and 40 km altitude.

| Altitude | 2 | 5 | 10 | 20 | 30 | 40 |
|---|---|---|---|---|---|---|
| U | 0.99999 | 0.99999 | 0.99999 | 0.99999 | 0.99999 | 0.99999 |
| V | 0.99999 | 0.99999 | 0.99999 | 0.99999 | 0.99999 | 0.99999 |
| W | 0.99999 | 0.99999 | 0.99999 | 0.99999 | 0.99999 | 0.99999 |
| T | 0.99997 | 0.99998 | 0.99995 | 0.99999 | 0.99999 | 0.99999 |
| $H_2O$ | 0.99999 | 0.99999 | 0.99999 | 0.99999 | 0.99999 | 0.99999 |
| $O_3$ | 0.99999 | 0.99999 | 0.99999 | 0.99999 | 0.99999 | 0.99999 |
| LWC | 0.99999 | 0.99999 | 0.99999 | - | - | - |
| IWC | 0.99999 | 0.99999 | 0.99999 | 0.99999 | - | - |

**Table A4.** Correlation coefficient for ZFP12 at 2, 5, 10, 20, 30 and 40 km altitude.

| Altitude | 2 | 5 | 10 | 20 | 30 | 40 |
|---|---|---|---|---|---|---|
| U | 0.99999 | 0.99999 | 0.99999 | 0.99999 | 0.99999 | 0.99999 |
| V | 0.99999 | 0.99999 | 0.99999 | 0.99999 | 0.99999 | 0.99999 |
| W | 0.99997 | 0.99997 | 0.99997 | 0.99998 | 0.99998 | 0.99999 |
| T | 0.99997 | 0.99998 | 0.99994 | 0.99998 | 0.99998 | 0.99998 |
| $H_2O$ | 0.99998 | 0.99997 | 0.99997 | 0.99976 | 0.99990 | 0.99989 |
| $O_3$ | 0.99995 | 0.99989 | 0.99998 | 0.99998 | 0.99997 | 0.99997 |
| LWC | 0.99996 | 0.99992 | 0.99969 | - | - | - |
| IWC | 0.99997 | 0.99996 | 0.99997 | 0.8917 | - | - |

**Table A5.** Correlation coefficient for ZFP8 at 2, 5, 10, 20, 30 and 40 km altitude.

| Altitude | 2 | 5 | 10 | 20 | 30 | 40 |
|---|---|---|---|---|---|---|
| U | 0.99922 | 0.99960 | 0.99968 | 0.99965 | 0.99973 | 0.99982 |
| V | 0.99916 | 0.99959 | 0.99959 | 0.99967 | 0.99958 | 0.99975 |
| W | 0.99443 | 0.99733 | 0.99733 | 0.99776 | 0.99780 | 0.99861 |
| T | 0.99997 | 0.99998 | 0.99998 | 0.99998 | 0.99998 | 0.99998 |
| $H_2O$ | 0.99859 | 0.99872 | 0.99872 | 0.98790 | 0.99243 | 0.98907 |
| $O_3$ | 0.99720 | 0.99499 | 0.99499 | 0.99936 | 0.99856 | 0.99828 |
| LWC | 0.99167 | 0.98857 | 0.98857 | - | - | - |
| IWC | 0.99559 | 0.99635 | 0.99635 | -0.08966 | - | - |



## 395 Appendix B: Trajectory Deviations

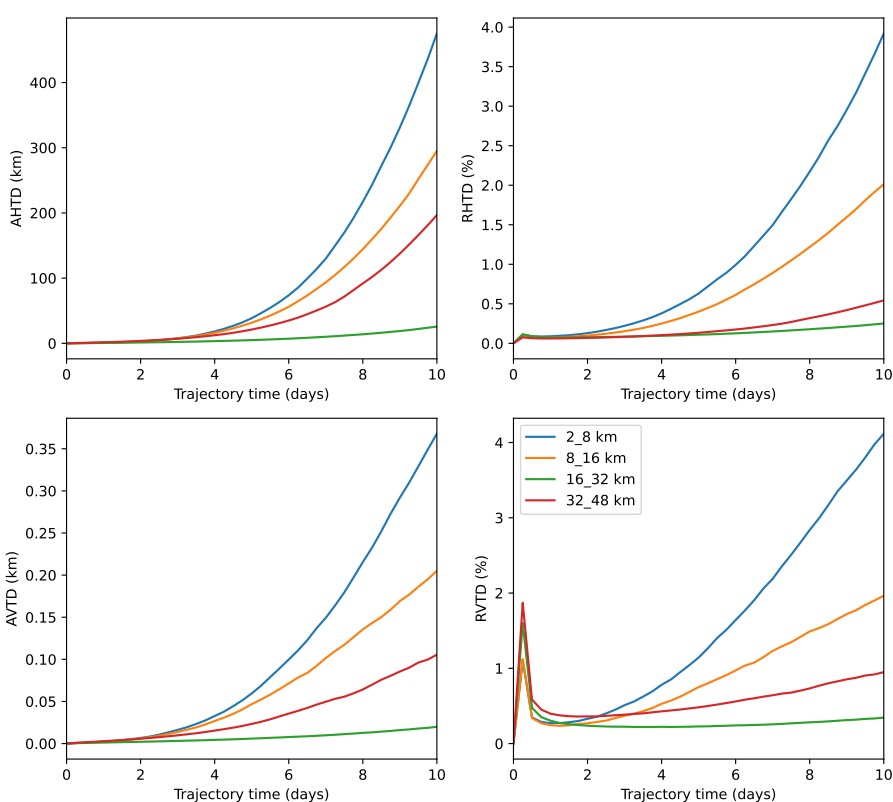

**Figure B1.** Same as Fig. 5 and Fig. 6 but for ZFP12.




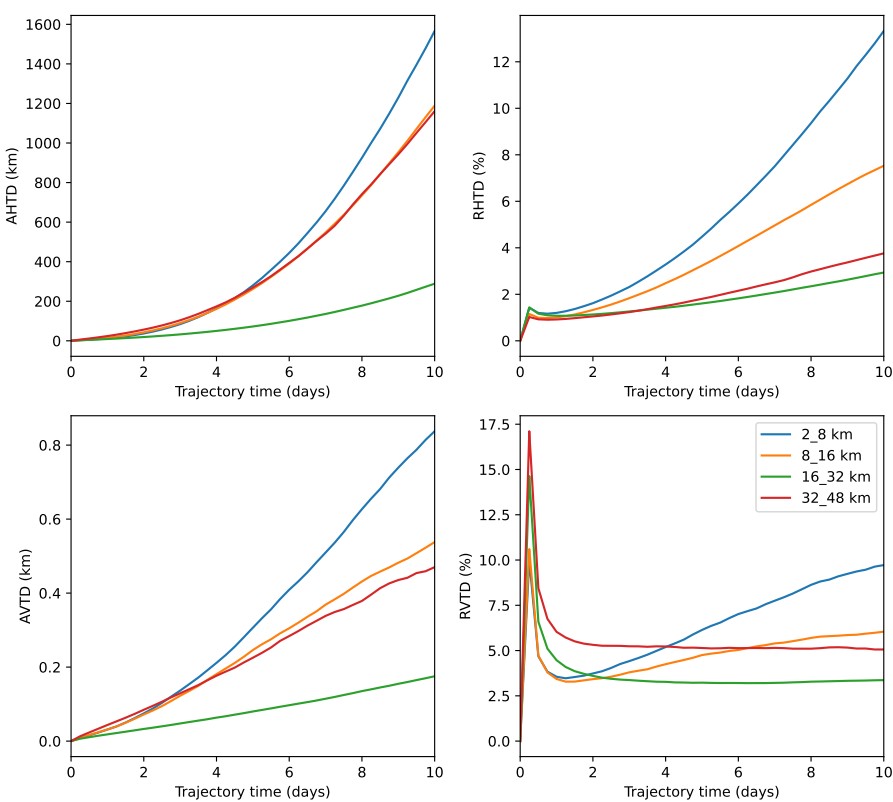

**Figure B2.** Same as Fig. 5-B1 but for ZFP8.



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
