# Peer review of "Compression of ERA5 meteorological reanalysis data and their application to simulations with the Lagrangian model for Massive Parallel Trajectory Calculations (MPTRAC v2.7)"

_EGUsphere, 2025_

## Author Comment (AC1)

**Reply to Referee 1**

**We thank referee 1 for the constructive, helpful criticism and the suggestion for revision. We have thoroughly revised the manuscript based on the comments given by the referees. A detailed point-by-point response to the comments by referee 1 are given below.**

The manuscript investigates the lossy compression of ERA5 reanalysis data and its impact on trajectory calculations using the Lagrangian model for Massive-Parallel Trajectory Calculations (MPTRAC). The topic is timely and relevant, with clear implications for data storage optimisation, I/O performance, and reproducibility in geoscience workflows. ERA5 is one of the most widely used reanalysis datasets in the atmospheric sciences, with applications ranging from climate research to operational forecasting, and it is increasingly important as a training and validation source for AI models. The authors assess two lossy compression methods (ZFP and Layer Packing, PCK) and one lossless compressor (ZSTD), examining their influence on 10-day forward trajectories distributed globally in the free troposphere and stratosphere.

The study addresses an important problem and contributes to the relatively underexplored area of quantifying the impact of lossy compression on scientific analyses. Work along these lines can help the community better understand these impacts and support the adoption of compression methods that offer substantial data reduction without compromising scientific integrity.

However, the current version omits substantial parts of the relevant literature and does not sufficiently engage with the state of the art in scientific lossy compression. Several methodological choices also weaken the strength of the conclusions. The following points require major attention:

1. Omission of relevant literature – Key works are missing, including Tinto et al. (2024, GMD, (https://doi.org/10.5194/gmd-17-8909-2024)), which also deals with the impact of lossy compression of geoscientific data, and other publications that define the current state of the art.

**Thanks a lot for this comment. We have added the work of Tinto (2024) as well as the works of Baker (2016); Dueben (2019); Delaunay (2019); Zender (2016) and Poppick (2020). These are cited in the introduction and the paragraph on P2, L59 has been changed as follows:**

**"Geoscientific data as e.g meteorological reanalysis data, climate simulation data and satellite data have increased immensely in size and their application in full resolution has become quite challenging for users. This is a known problem and has been in the focus of several previous studies. The efficiency of compressing climate simulation data was tested by e.g. Baker (2016); Dueben (2019) and Poppick (2020) and the compression of satellite data sets was tested by e.g. Delauney et al. (2019). Compression of meteorological reanalysis data was the focus of the study by Tinto (2024). In the geoscientific community the netCDF4 or HDF5 formats are widely used and thus compression of data sets in these formats was the focus of the studies by Delaunay (2019) and Zender (2016). All these studies came to the conclusion that lossy data compression is promising for reducing storage requirements. However, Poppick (2020) point out that it is important to evaluate the quality of compression in order to ensure that minimal scientific information is lost due to compression."**

2. Ignoring state-of-the-art compressors – The evaluation is limited to ZFP, PCK, and ZSTD,

omitting widely recognised high-performance compressors such as SZ and MGARD. Without including these methods, the results cannot be considered representative of current capabilities of lossy compression for ERA5 data compression.

**We agree that including also SZ and MGARD would be quite valuable, however this is beyond the scope of this study. Our intention for this study is not to test which of all available compression methods is the most efficient compressor for the ERA5 data, but to understand what impact the compression of the meteorological reanalysis data has on the trajectory calculations with MPTRAC. Further, to our knowledge, previous studies have also typically focused on a selection of compression methods and setups, rather than attempting to exhaustively compare all possible options. We agree that some of our comments made concerning PCK were inadequate since we are not considering the full range of possible compression techniques available and set-ups of the respective techniques that are possible. We will remove the misleading statements we made.**

3. Suboptimal use of ZFP – ZFP is applied in precision mode, which the literature reports as less efficient and with poorer rate–distortion performance than accuracy mode. This disadvantages ZFP in the comparisons and may bias the conclusions.

**We thank the reviewer for this important comment. We acknowledge that there are studies reporting better rate–distortion performance for ZFP in accuracy (absolute tolerance) mode compared to precision mode. We agree that the optimal choice of mode and error bounds may depend strongly on the variable considered: for parameters that vary by orders of magnitude with altitude, relative precision can be advantageous, whereas for parameters that remain within a similar order of magnitude across levels, absolute tolerance may provide better compression efficiency. A systematic evaluation of these options would indeed be valuable, but it is unfortunately beyond the scope of the present work. Our study builds on a setup for ZFP that was previously tested within our group and found to perform reasonably well for ERA5 data, and our focus here is on understanding the implications of compression for trajectory simulations rather than on identifying the optimal ZFP configuration.**

4. Unsupported claim that PCK is the "best choice" – The conclusion that PCK is the most suitable compressor lacks supporting evidence, as state-of-the-art methods are not included and ZFP is used in a suboptimal configuration. This risks misleading readers about PCK's competitiveness.

**We totally agree and we once again would like to apologize for our misleading comments. For us it was the best choice because it is easy to apply since no user specific set-ups are needed and it results in a 50% reduction of the files with additionally maintaining a high accuracy for the trajectory calculations. In addition, the PCK method offers the advantage of extremely fast decompression, requiring only the application of scaling and offset factors, and significantly reduced runtime needed for file input. Overall, it was the only method among those tested that reduced both file size and runtime requirements simultaneously. We make this point much more clear now in the manuscript and have checked our manuscript for misleading statements and removed/revised these. E.g. a misleading statement was made in the abstract. The sentence there has been changed to:**

**"Thus, our study shows that all compression methods considered here (ZSTD, PCK and**

ZFP) would be valuable for application in atmospheric sciences and that with compression of the ERA5 meteorological reanalyses data one can overcome the challenges of high demand of disk space from this data set."

We would like to keep the sentences concerning PCK in the discussion and conclusion since these refer to our personal choices we made concerning compression of ERA5 data and MPTRAC trajectory calculations. However, we adjusted these as follows to be more clear:

"For us, PCK is the best choice, since it is also for inexperienced users easy to apply and is quite efficient on our current supercomputer system at compressing the ERA5 data (CR=2) while at the same time keeping the accuracy of the data, resulting in low transport deviations ($< 40$ km). ZSTD has the advantage of being a lossless compression method, so there is no loss, and storage requirements are reduced by 30%. ZFP has the advantage of allowing users to specify themselves the level of compression."

"For our needs PCK is the best choice and has thus been implemented in MPTRAC as the default compression method. However, ZFP and ZSTD can be also used by enabling them when compiling MPTRAC. In the future, we plan to add other widely used compression methods, such as SZ3 and MGARD."

Recommendation: I recommend major revisions. The authors should (1) expand the literature review to include key recent works and provide proper context on the state of the art, (2) revisit the ZFP configuration to use competitive modes reported in the literature, and (3) either include tests with state-of-the-art compressors such as SZ or explicitly limit their claims, providing a clear justification for the exclusion of these methods.

**Thanks again for your valuable comments. We have taken all your comments into account and hope that the revised version covers now all these points to your satisfaction**.

**References**

Baker, A. H., Hammerling, D. M., Mickelson, S. A., Xu, H., Stolpe, M. B., Naveau, P., Sanderson, B., Ebert-Uphoff, I., Samarasinghe, S., De Simone, F., Carbone, F., Gencarelli, C. N., Dennis, J. M., Kay, J. E., and Lindstrom, P.: Evaluating lossy data compression on climate simulation data within a large ensemble, Geosci. Model Dev., 9, 4381–4403, https://doi.org/10.5194/gmd-9-4381-2016, 2016.

Delaunay, X., Courtois, A., and Gouillon, F.: Evaluation of lossless and lossy algorithms for the compression of scientific datasets in netCDF-4 or HDF5 files, https://doi.org/10.5194/gmd-12-4099-2019, 2019.

Düben, P. D., Leutbecher, M., and Bauer, P.: New Methods for Data Storage of Model Output from Ensemble Simulations, Mon. Weather Rev., 147, 677–689, https://doi.org/10.1175/mwr-d-18-0170.1, 2019.

Poppick, A., Nardi, J., Feldmann, N., Baker, A., Pinard, A., and Hammerling, D. M.: A statistical analysis of lossily compressed climate model data, Comput. Geosc., 145, 104 599, https://doi.org/10.1016/j.cageo.2020.104599, 2020.

Tintó Prims, O., Redl, R., Rautenhaus, M., Selz, T., Matsunobu, T., Modali, K. R., and Craig, G.: The effect of lossy compression of numerical weather prediction data on data analysis: a case study using enstools-compression 2023.11, Geosci. Model Dev., 17, 8909–8925, https://doi.org/10.5194/gmd-17-8909-2024, 2024.

Zender, C. S.: Bit Grooming: statistically accurate precision-preserving quantization with compression, evaluated in the netCDF Operators (NCO, v4.4.8+), Geosci. Model Dev., 9, 3199–3211, https://doi.org/10.5194/gmd-9-3199-2016, 2016.

---

## Author Comment (AC2)

**Reply to Referee 2**

**We thank referee 2 for the constructive, helpful criticism and the suggestion for revision. We have thoroughly revised the manuscript based on the comments given by the referees. A detailed point-by-point response to the comments by referee 2 is given below.**

This study evaluated various compression models at different compression rates applied to ERA5 reanalysis data. The compressed ERA5 output was then used to drive MPTRAC for trajectory calculations. The paper investigates how both the choice of compression scheme and compression rate affect the accuracy of trajectories computed by MPTRAC.

The manuscript is well-structured overall, but the writing could benefit from some revision for clarity and flow, as it's currently a bit difficult to read. Below are my comments:

Major Comments

Fig 5-Fig6

- Do you have an explanation for why the deviation is larger near the surface.

  **The stratosphere is more stable which suppresses vertical mixing. Therefore, in the stratosphere, air parcels follow smoother, more predictable paths. In contrast, the troposphere is more unstable and thus affected by turbulence and convection which makes trajectories harder to predict over time. Thus, it is not astonishing that in the lowest considered altitude range (2-8 km) the largest deviations are found.**

  **We have added the following sentence in Sect. 3.2.1 to explain this: "This is because the stratosphere is stable stratified and the wind fields are much smoother, allowing for more accurate trajectory calculations. In contrast, the troposphere is much more dynamically variable with turbulence, convection and vertical mixing, making accurate trajectory calculation more challenging."**

- I am curious to know why the deviation is negligible in the days < 4 regardless of the compression method and compression level? Why does the deviation grow in time? The similar think is also observed in Fig 7 as well.

  **Trajectory calculations suffer from several error sources and the errors accumulate and amplify over time (Kuo, 1985; Stohl, 1998; Harris, 2005; Engström, 2009). The longer the simulation or the more steps are taken, the more these errors can cause deviations from the true trajectory. This is why the deviations get larger as longer the trajectories are calculated.**

  **We have added the following sentence to Sect. 3.2.1: "This development in time is typical for transport deviations, since trajectory calculations suffer from several errors that accumulate over time (e.g. Kuo, 1985; Stohl, 1998; Harris, 2005; Engström, 2009)"**

Minor Comments

Line 149-151: Is there a reason for choosing the tolerance option for geopotential height and Temperature while choosing the precision mode for all other variables?

**The set-up for ZFP that we use was derived in previous studies made by one of our former group members and was found to work reasonable well with this data set.**

- Line 224-226: It was mentioned that the correlation coefficient reached 0.99999 when the precision mode was used. What's the reason for choosing the tolerance option for T?

**Although we could achieve a higher correlation coefficient in this test case, we kept our original set-up. This is because (1) we tested this only for ZFP16 and not for ZFP12 and ZFP8 and (2) it did not change the results concerning the trajectory calculations.**

Line 233-235: According to Table 1, file size after the compression by PCK is larger than the file size after the compression by ZFP. Why do you think reading input file takes the shortest time for PCK and not ZFP compressed files?

**This is because of the way how the data is compressed and then again decompressed. PCK was specifically designed for netCDF files and stores the data as arrays of scale and offset values. ZFP on the other hand comes from the image processing and devides the data into fixed size blocks of dimensions 4 × 4 × 4 that are each stored using the same, userspecified number of bits. Although the compression ratios are higher for ZFP and the file sizes are smaller, the decompression is faster with PCK.**

Fig4: Lines are labeled as physics, input, output, and total. But the caption or the main text is missing the explanations for those labels.

**We have added the following sentence to the figure caption and to the main text to explain the used labels.**

**"Timers were used for reading the input data (Input), performing the trajectory calculation (Physics), writing the trajectory data to an ascii or netcdf file (Output) and the total time spent for the trajectory calculation (Total)."**

I would suggest labelling each panel in each figure.

**Thanks for the suggestion. We have labelled all figures. Additionally we have, due to a comment by the Editorial Support Team that came up during file validation, added to the linestyle markers, so that the figures are readable for readers suffering from all kinds of coulour blindness.**

Is Fig5-6 are ensemble mean? Or from a single trajectory?

**This is a mean over all globally distributed $10^6$ trajectories that we have calculated. To make this more clear to the reader we have rewritten the sentence in Sect. 3.2.1 and refer also to Sect. 2.5 where the equation and some further explanations are given.**

**"Figures 5 and 6 and Fig. B1 and B2 show the absolute and relative horizontal and vertical transport deviations (AHTD, RHTD, AVTD and RHTD, respectively) for the $10^6$ 10-day trajectories (mean over all trajectories, see Sect. 2.5) that have been calculated with MPTRAC using the compressed reanalysis data files (PCK, ZFP16, ZFP12 and ZFP8)"**

Line 285-287: Do you mean that the maximum deviation frequencies are similar between the

trajectories started at tropospheric and stratospheric altitudes?

**Yes, for ZFP8 the mximum deviation frequencies for trajectories started in the strato-sphere are almost as high as the ones started at tropospheric altitudes. We changed the sentence as follows to be more clear:**

**"For the trajectories calculated based on the ZFP8 compressed data the maximum deviations for the trajectories started at stratospheric altitudes occur almost as frequent as for the ones started at tropospheric altitudes."**

**References**

Engström, A. and Magnusson, L.: Estimating trajectory uncertainties due to flow dependent errors in the atmospheric analysis, Atmos. Chem. Phys., 9, 8857–8867, 10.5194/acp-9-8857-2009, 2009.

Harris, J. M., Drexler, R. R., and Oltmans, S. J.: Trajectory model sensitivity to differences in input data and vertical transport method, J. Geophys. Res., 110, D14109, 10.1029/2004JD005750, 2005.

Kuo, Y.-H., Skumanich, M., Haagenson, P. L., and Chang, J. S.: The accuracy of trajectory models as revealed by the observing system simulation experiments, Mon. Weather Rev., 113, 1852–1867, 10.1175/1520-0493(1985)113¡1852:TAOTMA¿2.0.CO;2, 1985.

Stohl, A.: Computation, accuracy and applications of trajectories—a review and bibliography. Atmospheric Environment, 32, 947–966, 10.1016/S1352-2310(97)00457-3, 1998.

---

## Author Comment (AC3)

**Reply to Editor**

To aid the authors with producing a manuscript revision which will quickly be accepted, I'd like to point out that I found responses to reviewer 1's major concerns 2 and 3 to be insufficient. The goal of GMD is to provide readers with an understanding of *cutting-edge* innovations in geophysics. Evaluating against old/suboptimal implementations of rival schemes is at best not state-of-the-art and at worst misleading. I'm sympathetic with the fact that adding new compression methods would be a lot of work, but I will need to see much better justification for keeping things as-is and/or big changes to the manuscript to feel comfortable publishing this work without including more modern compression methods in the evaluation.

**We would like to thank the editor for these comments. We understand the concerns raised and would like to clarify several points. First, we neither use any old or suboptimal implementations of the compression methods applied here. The compression methods used here (ZFP, ZSTD and PCK) are all up to date and widely used by the scientific computing and climate communities. Thus, the results presented here are valid and not misleading. SZ3 and MGARD are not more modern compression methods than the here used compression methods ZFP, PCK and ZSTD.**

**Second, ZFP has continously undergone further developed since its first release (Lindstrom et al., 2025). ZFP is a very powerful compression method that offers several modes (fixed accuracy, fixed precision, fixed rate and an "expert" mode). Which mode to be used can be chosen by the user and there are, to our knowledge, no studies that state that one of the modes is not suitable or performing less then the other mode. Diffenderfer et al. (2019) provide a detailed error analysis for ZFP, clearly stating that the errors for all modes are similar. The same holds for studies where ZFP was compared with other compression methods as e.g. SZ3. The only study that may indicate that the accuracy mode could perform better than the precision mode, is the study from Tinto Prims et al. (2024). This result can only be inferred by examining their figures. However, the paper does not explicitly disucss which mode performs better. Furthermore, Tinto Prims et al. (2024) do not provide the error bounds, so we cannot repeat their compression test. Additionally, the used ERA5 test data may differ. Each user downloads different subsets of meteorological parameters with different resolutions. Even knowing the error bounds of Tinto Prims et al. (2024), the results would be most probably not comparable.**

**Third, we would like to emphasize once again that our study does not focus on determining the best compression method for the ERA5 data. Our goal is to rather understand how using compressed data files as input files affect trajectory calculations. Testing all compression methods and set-ups is impossible since there are too many options. Previous studies also selected a limited number of set-ups and methods to test (e.g. Baker et al., 2019; Alfarov et al., 2019; Delaunay et al., 2019; Poppick et al., 2020; Huang and Hoefler, 2023; Baker et al., 2024; Tinto Prims et al., 2024).**

**References**

**Alforov, Y., A. Novikovay, M. Kuhny, J. Kunkel and T. Ludwig: Towards Green Scientific Data Compression Through High-Level I/O Interfaces, 30th International Sym-**

posium on Computer Architecture and High Performance Computing (SBAC-PAD), https://doi.org/10.1109/CAHPC.2018.8645921, 2019.

Baker, A. H., Hammerlin, D.M. and Turton, T. L.: Evaluating image quality measures to assess the mpact of lossy compression applied to climate simualation data, Eurographics Conference on Visualization (EuroVis), 38, 3, https://doi.org/10.1111/cgf.13707, 2019.

Baker, A. H., Pinard, A., and Hammerling, D.: On a strcutrual similarity index approach for floating-point data, IEEE Transactions on Visualization and Computer Graphics, 30, 9, https://doi.org/10.1109/TVCG.2023.3332843, 2024.

Huang, L. and Hoefler, T.: Compressing multidimensional weather and cliamte data into neural networks, Published as a conference paper at ICLR 2023, published on ArXiv, https://doi.org/10.48550/arXiv.2210.12538, 2023.

Delaunay, X., Courtois, A. and Gouillon, F.: Evaluation of lossless and lossy algorithms for the compression of scientific datasets in netCDF-4 or HDF5 files, https://doi.org/10.5194/gmd-12-4099-2019, 2019.

Diffenderfer, J. D. , Fox, A. L., Hittinger, J. A. F., Sanders, G. and Lindstrom, P.: Error analysis of ZFP compression for floating-point data, SIAM Journal on Scientific Computing, 41, A1867–A1898, https://doi.org/10.1137/18M1168832, 2019.

Lindstrom, P., Hittinger, J., Diffenderfer, J., Foy, A., Osei-Kuffuor and Banks, J.: ZFP: A compressed arrray representation for numerical computations, Int. J. High Perform. Comp. Appl., 39, 1, 104-122, https://doi.org/10.1177/10943420241284023, 2025.

Tintó Prims, O., Redl, R., Rautenhaus, M., Selz, T., Matsunobu, T., Modali, K. R., and Craig, G.: The effect of lossy compression of numerical weather prediction data on data analysis: a case study using enstools-compression 2023.11, Geosci. Model Dev., 17, 8909–8925, https://doi.org/10.5194/gmd-17-8909-2024, 2024.

Poppick, A., Nardi, J., Feldmann, N., Baker, A., Pinard, A., and Hammerling, D. M.: A statistical analysis of lossily compressed climate model data, Comput. Geosc., 145, 104 599, https://doi.org/10.1016/j.cageo.2020.104599, 2020.